# Stress-buffering effects of physical activity and cardiorespiratory fitness on metabolic syndrome: A prospective study in police officers

**René Schilling**⊙*, **Flora Colledge, Uwe Pühse, Markus Gerber**

Department of Sport, Exercise and Health, University of Basel, Basel, Switzerland

* rene.schilling@unibas.ch

## Abstract

Metabolic syndrome (MetS) is a worldwide health concern related to cardiovascular disease. Stress at work increases the risk for MetS, whereas physical activity and cardiorespiratory fitness (CF) have been shown to be potential buffers against stress. The aim of this study was to test the stress-buffering effects of physical activity and CF on the relationship between work stress and MetS. In a prospective study, we followed 97 police officers (mean age = 39.7 years; mean body mass index = 25.74 kg/m$^2$) over one year and assessed MetS, as defined by the National Cholesterol Education Program Adult Treatment Panel III. Stress at work was measured with the Job Content Questionnaire, as well as the Effort-Reward Imbalance Questionnaire. Physical activity was assessed objectively via 7-day accelerometry. CF was assessed with the Åstrand bicycle ergometer test. Hierarchical linear regression models were carried out to predict MetS at follow-up (mean overall MetS score = 1.22), after controlling for baseline levels and sociodemographic background (mean overall MetS score = 1.19). Higher CF levels were significantly associated with lower MetS risk at follow-up (β = -.38). By contrast, no main effects were found for physical activity and work stress. However, high effort and demand were significantly correlated with increased blood pressure (effort: $r$ = .23 for systolic blood pressure; $r$ = .21 for diastolic blood pressure) and waist circumference (effort: $r$ = .26; demand: $r$ = .23). Moreover, no significant interaction effects occurred between work stress and CF/physical activity. The results emphasize the importance of high levels of CF in the prevention of MetS in police officers. Accordingly, provision of regular training opportunities and repeated CF testing should be considered as a strategy in overall corporate health promotion.

## Introduction

Metabolic syndrome (MetS) has become a prominent health concern in industrialized countries [1, 2]. MetS is a cluster of symptoms, consisting of abdominal adiposity, reduced glucose tolerance, dyslipidemia, and hypertension [1]. While accepted and clinically used definitions

**Data Availability Statement:** The datasets used and/or analysed during the current study are available on reasonable request from Ms. Nienke Jones (Nienke.jones@bs.ch; +41 61 268 13 54) via the Ethics Committee of Northwestern and Central

Switzerland (EKNZ). At the time of obtaining ethical clearance for the present study from the EKNZ, and in line with Swiss laws, we stated that only authorized researchers who are directly involved in the present project will have access to the raw data. Accordingly, and in line with this statement, we cannot grant access to the data for third parties, unless this is officially approved by the EKNZ.

**Funding:** The authors received no specific funding for this work.

**Competing interests:** The authors have declared that no competing interests exist.

differ slightly, scholars generally agree on strong associations between MetS and cardiovascular diseases [1, 3]. Widely accepted criteria for MetS were established by the National Cholesterol Education Program [4]. The systematic review and meta-analysis by Mottillo et al. [5] concluded that meeting these criteria was linked to a 100 percent increased risk for cardiovascular mortality.

A chronically elevated level of stress is regarded as a salient risk factor for MetS [6–8]. In Western societies, the place of work constitutes a major source of stress for many adults [9]. Two theoretical work stress models dominate current stress research [10]. Firstly, the job strain model, which holds that stress arises from a discrepancy between demands and control [11]. Secondly, the effort-reward imbalance model, which assumes that stress is due to a mismatch between commitment and gratification [12]. Both models have shown that increased stress is associated with a heightened risk of MetS [13–15]. For example, Chandola et al. [14] revealed the risk for MetS is twice as high in participants who report high vs. low levels of job strain over the span of 14 years.

Police officers encounter a multitude of psychosocial stressors during their work [16]. In a study with American police officers, Violanti et al. [17] showed that night shifts, fewer sleeping hours and overtime might contribute to the development of MetS in police officers. However, police officers form a heterogeneous population, and it is suggested that individual stress perceptions play a more important role than their specific divisional tasks [18]. For example, Gerber et al. [19] reported closer links between police officers' health and subjective stress perception than for shift work status. Similarly, Garbarino and Magnavita [20] followed police officers for 5 years, and found that high rates of work stress (job strain and effort-reward imbalance) were associated with a 2.7 times higher risk for the development of MetS. Prevalence rates of MetS are rising [21–23], and police officers have been found to be at increased risk compared to the general population [24]. Despite these findings, however, prospective studies examining the association between work stress and MetS are still relatively rare [15].

Regular physical activity and resulting cardiorespiratory fitness (CF) have been shown to enhance resilience against stress and stress-related disease [25–29], including police officers [30, 31]. However, with a few exceptions [27], most of these studies used self-reported health indicators, and none have focused on MetS. Therefore, the aim of the present study is to investigate potentially stress-buffering effects of CF on MetS. The study uses a 1-year prospective design to assess the effects of work stress on the concurrent and future level of MetS. As noted above, the occupation of policing is known to involve a stressful work environment and an increased risk for MetS [24, 32, 33].

Based on the literature presented above, three hypotheses will be tested in the present paper: First, high levels of CF and physical activity are negatively associated with MetS [34]. Second, high levels of work stress are positively associated with MetS [35]. Third, CF and physical activity moderate (buffer) the association between occupational stress and MetS. In other words, among police officers with high stress levels, those who are more physically active or fit are less likely to have or develop MetS [26].

## Methods

### Study design, participants and procedures

The present study was designed as a 12-month prospective investigation with two data assessments. The present sample consisted of police officers from Basel, Switzerland. The recruitment involved different workplace dissemination channels, consisting of videos, news journals, flyers, and information during team meetings. These advertisements promoted a comprehensive health check (including MetS), work stress assessment, cardiorespiratory

fitness test, and 7-day actigraphy, all of which were performed twice within one year. Anyone interested was invited to an e-learning program that provided information about the study background, measurements, risks, and benefits. Participation was voluntary without financial incentives. However, all activities related to the study could be performed during working hours, and participants received a personalized health profile. Furthermore, a voluntary life-style-coaching was offered to all participants. Prior to the health check, participants provided written informed consent and confirmed their physical eligibility based on the Physical Activity Readiness Questionnaire (ACSM) [36]. In case of uncertainty, a general practitioner was consulted. If (moderate-to-vigorous-intensity) physical activity was contraindicated, only non-physical measurements were performed. All study procedures followed the principles of the Declaration of Helsinki; the study was approved by the local ethics committee Ethikkommission Nordwest- und Zentralschweiz (EKNZ: Project-ID: 2017–01477).

The health and fitness checks took place between October 2017 and April 2019, in a laboratory at the education and training center of the police force. The specific room was exclusively reserved for the study across the entire study period. The laboratory sessions lasted approximately 60 minutes. On the same day, participants answered an online questionnaire addressing their socio-demographic background and their occupational stress level. Furthermore, participants answered questionnaires on burnout symptoms, overall mental health, and sleep complaints (not discussed in this paper). Following the laboratory session, participants were asked to wear an accelerometer for seven consecutive days to measure physical activity. An additional smartphone-based real-life measurement of emotions and stressors was performed during the first two days after the laboratory session; this data will be presented elsewhere.

## Measures

**Metabolic syndrome (MetS).**   Our criteria for MetS followed the recommendations of the National Cholesterol Education Program Adult Treatment Panel III (NCEP III): (i) waist circumference > 102 for men, and > 88 cm for women; (ii) triglyceride level > 1.7 mmol/l; (iii) HDL cholesterol of < 1.0 mmol/l for men and < 1.3 mmol/l for women; (iv) systolic blood pressure ≥ 130 mmHg or diastolic blood pressure ≥ 85 mmHg; and (v) blood HbA1c level > 6.1 percent [4]. Based on whether these cut-off values were exceeded (score = 1) or not (score = 0), we calculated an overall MetS sum score for further analyses. NCEP III describes the presence of three or more of the risk determinants as MetS.

In our study, the assessment of each MetS component was performed as follows: We assessed waist circumference following the World Health Organization (WHO) STEPS Manual [37]. Participants were asked to stand and breath in a relaxed way. A tape measure was applied in horizontal position in the middle of the inferior margin of the lowest rib and the crest of the ilium. The measurement was taken at the end of the expiration. Blood samples were drawn via finger prick following WHO guidelines [38]. The sample was instantly analyzed with Alere Afinion AS100 Analyzer (Abbott Diagnostics, Alere Technologies, Rodeløkka NO-0504 Oslo, Norway). HbA1c, high-density lipoprotein cholesterol (HDL-C), and triglycerides (TG) were assessed for further analysis. Validity and reliability have been described previously [39]. Lipid and glucose panel controls were frequently used to ensure the reliability of the devices. Systolic and diastolic blood pressure were measured in a sitting position with the OMRON M500 (OMRON Healthcare Co. Ltd. 53, Kunotsubo, Terado-cho, Muko, Kyoto 617–0002 Japan). The device was attached to the left arm. Two measurements were taken, spaced apart by 3 minutes. Systolic and diastolic blood pressure were noted and the mean was calculated for further analyses.

**Work stress.** Work stress was assessed using the Job Content Questionnaire (JCQ) [40] as well as the Effort-Reward Imbalance (ERI) Questionnaire [12]. The JCQ consists of a demand and a control scale. The demands were assessed with five items (e.g. 'My job requires working very hard.'), and answers were given on a 4-point Likert scale (1 = never to 4 = often). The control subscale consisted of six items (e.g. 'I have freedom to make decisions about my job.') with the same response options as for the demands subscale. The items were summed for each subscale and a job demand and control ratio (JDC ratio) was computed using the formula: demand / control * 0.8333. Values above 1 indicate work stress with possible negative effects on health [40]. The validity and reliability of the JCQ has been described previously [41]. The effort scale of the ERI questionnaire consists of five items (e.g. 'I have a lot responsibility in my job.'), whereas the reward scale comprises eleven items (e.g. 'Considering all my efforts and achievements, my job promotions are adequate.'). All items were answered on a 5-point Likert scale from 1 (none) to 5 (very high). Each subscale was summed and an overall ratio was calculated with the formula: effort / reward * 0.4545. An ERI ratio higher than one has been typified as high work stress [12]. Reliability and validity of both job stress questionnaires have been described previously [41, 42]. Reviews of prospective cohort studies have shown an increased risk for CVD in individuals with high work stress, as measured with the JCQ or the ERI questionnaire [43, 44].

**Physical activity.** Physical activity was assessed objectively with accelerometry. Accelerometry was carried out using ecgMove3 sensors (movisens GmbH, Karlsruhe, Germany). The ecgMove3 records 3-dimensional acceleration (63 Hz) and barometric altitude (8 Hz). Evidence for the validity and reliability has been provided previously [45, 46]. At the end of the laboratory session, participants were asked to put on the device, which was worn on a textile dry electrode chest belt. Data was saved on an internal memory and readout after the device was returned. Data was processed with the DataAnalyzer (movisens GmbH, Karlsruhe, Germany). The software provides a report of energy expenditure, steps, activity classes, and non-wear time. Following Troiano et al. [47], days with $\geq$ 10 hours of wear time were considered valid. Datasets with $\geq$ 5 valid days were included in the data analysis. The average day values (minutes per valid day) of moderate-to-vigorous physical activity (MVPA) was used for further calculations.

**Cardiorespiratory fitness (CF).** CF was assessed with the widely used Åstrand submaximal bicycle test [48]. The test accurately estimates maximal oxygen consumption (VO$_2$max) based on standardized extrapolations of heart rates at certain resistances [48–50]. Participants were instructed to avoid any strenuous activity 24 hours prior to the testing. Furthermore, no meals and liquids were to be consumed within three hours prior to the testing. For the test, participants wore a POLAR (Polar Electro, Kempele, Finland) heart rate monitor. Standardized starting workloads (men = 150 Watts; women = 100 Watts) were adjusted so that the heart rate remains in predefined limits. These limits were 130–160 bpm for participants < 40 years of age, and 120–150 (bpm) for participants $\geq$ 40 years. Cycling cadence was set at 60 rotations per minute. Borg ratings were assessed after every minute and participants were controlled for cancellation criteria [51]. After six minutes of cycling, the test ended. Participants were only asked to proceed for another minute if the heart rate of the last two minutes varied by more than 5 beats per minute. The resulting final workload and mean heart rate of the last two minutes were translated into age and gender adjusted VO$_2$max (ml/kg/min) levels. Following the American College of Sports Medicine (ACSM), participants' CF were classified as 'very poor', 'poor', 'fair', 'good', 'excellent', and 'superior' [51].

## Statistical analyses

Descriptive statistics were calculated for all main study variables at baseline and additionally for the various MetS components at follow-up (data on physical PA and CF at follow-up was not considered in the present data analysis). Furthermore, cross-sectional bivariate correlations are provided for physical activity (accelerometry), CF, and occupational stress (JDC ratio, ERI ratio), with the MetS overall score at baseline and follow-up. To explore possible stress-buffering effects of physical activity and CF on MetS, we provide the results of several hierarchical linear regressions. Following procedures by Aiken and West [52], stress-buffering effects were determined as significant interaction terms in a moderation model. Separate moderation models were tested for objectively assessed physical activity, as well as for CF, and their respective interactions with the two work stress indicators. In order to find out which variables were initially associated, we first performed cross-sectional (baseline) analyses. We then computed prospective analyses, accounting for the baseline level of MetS, in order to establish temporal precedence.

For cross-sectional analyses, we performed 4-step hierarchical linear regressions, based on participants' baseline scores. The variables were entered in the regression equation in the following order: socio-demographic background (step 1), work stress (step 2), physical activity (or CF) (step 3), interaction terms between JDC ratio and ERI ratio, physical activity (or CF) and JDC ratio, and physical activity (or CF) and ERI ratio (step 4). All variables were centered (z-standardized) before the interaction terms were calculated.

For the prospective analyses (with MetS at follow-up as dependent variable), we performed a series of 5-step hierarchical linear regression analyses. The variables were entered in the regression equation in the following order: socio-demographic background (step 1), baseline MetS score (step 2), work stress (step 3), physical activity (or CF) (step 4), interaction terms between JDC ratio and ERI ratio, physical activity (or CF) and JDC ratio, and physical activity (or CF) and ERI ratio (step 5). Again, all variables were centered (z-standardized) before the interaction terms were calculated. In the results section, we report the stepwise changes in $R^2$, and the standardized regression weights for each predictor variable in the final models. All statistical analyses were performed using SPSS 26 (IBM Corporation, Armonk NY, USA), and $p$-values of $< .05$ were considered as statistically significant.

# Results

## Sample description

Approximately 1000 police force employees received the study advertisement, which they could view voluntarily. From these, 227 officers (approximately 23%) agreed to obtain background information via the e-learning program, and 201 officers finally decided to participate in the cross-sectional study (88%). Of these, 97 (48.3%) officers also took part in the follow-up data assessment and were considered for data analyses. Among those participants with complete baseline and follow-up data, the mean age was 39.7 years (± 9.59), and the mean body mass index was 25.74 kg/m$^2$ (± 3.68). This sample was composed of 32 women (33%) and 65 men (67%). In sum, 74.2 percent (n = 72) of the participants reported being married or in a relationship, 11.3 percent (n = 11) had higher education (university or college), 34.0 percent (n = 33) had completed high school, and 48.5 percent (n = 47) had basic vocational training. Almost half of the participants (43.3%, n = 42) reported having children living at home, whereas 2.1 percent (n = 2) had current care responsibilities. In total, 44.3 percent (n = 43) of participants were involved in shift work and 15.6 percent (n = 15) were employed in middle and upper ranks of the department. The average employment rate was 80.5 percent, and 13.4

percent (n = 13) of the participants reported current intake of medication. Current smoking was reported by 19.6 percent (n = 19), weekly intake of alcoholic drinks was reported by 63.9 percent (n = 62). Finally, 48.5 percent (n = 47) of the participants stated that they do not think that they are physically active enough to maintain good health. As shown in Table 1, at baseline, participants who were lost until follow-up did not differ significantly from participants who participated in the follow-up data assessment in any of the sociodemographic background variables or the predictor and outcome variables. In the present paper, only data is considered from those 97 participants who took part in the follow-up data assessment, even when cross-sectional associations are reported.

## Descriptive statistics and bivariate correlations

Descriptive statistics for the participants who took part in the follow-up data assessment (N = 97) are presented in Table 2. At baseline, 7.2 percent (n = 7) of the participants had $\geq 3$ out of 5 criteria for MetS, whereas 8.3 percent (n = 8) fulfilled $\geq 3$ criteria at follow-up. With regard to work stress at baseline, more than half of the participants (51.5%, n = 50) had a JDC ratio $\geq 1$, whereas 44.3 percent (n = 43) had an ERI ratio $\geq 1$. Based on the accelerometry data, the present sample was highly active. The mean levels in the present sample appeared to be approximately 3 to 7 times higher than the WHO recommendations of $\geq 150$ minutes of weekly moderate-to-vigorous physical activity [53]. Furthermore, the classification of CF levels revealed that 25.8 percent (n = 25) had poor to very poor fitness, 24.8 percent (n = 24) showed fair to good fitness, whereas 48.5 percent (n = 47) reached the level of excellent to superior fitness.

Table 3 shows the correlations between physical activity, CF, and work stress with the components of MetS (both at baseline and follow-up). A significant and negative correlation was found between objectively assessed physical activity and blood sugar, both at baseline and follow-up. The negative correlation between physical activity and the overall score of MetS only reached significance for the follow-up measurement. Higher CF was significantly associated with more favorable scores in most MetS components. More precisely, CF was significantly and negatively correlated with waist circumference, blood lipids, blood sugar, and the overall MetS score at baseline; and with waist circumference, blood lipids, diastolic blood pressure, and the overall MetS score at follow-up.

Neither the JDC ratio, nor the ERI ratio, were significantly associated with the various components of MetS, at baseline or at follow-up. The demand subscale was only significantly associated with waist circumference at follow-up. Interestingly, the control subscale was positively associated with several of the MetS outcomes and reached the level of significance for diastolic blood pressure at follow-up. The effort subscale was significantly and positively related to systolic and diastolic blood pressure at baseline, whereas significant (positive) correlations with waist circumference and systolic and diastolic blood pressure were observed at follow-up. The reward subscale was not significantly associated with any of the outcomes.

## Cross-sectional and prospective hierarchical linear regressions

The results of the hierarchical linear regression analyses are reported in Tables 4 and 5. As shown in Table 4, the cross-sectional regression analyses explained between 18 and 30 percent in the dependent variable (overall MetS score). Nevertheless, the analyses revealed significant interaction effects only between the two job stress questionnaires, whereas no stress-buffering effects of physical activity or CF appeared to be present in the data. Education level was significantly and negatively associated with the overall MetS score. Age was positively associated with the overall MetS score; however, this association did not reach significance in the final

**Table 1. Differences between participants who were lost to follow-up and participants who completed both data assessments in sociodemographic background and predictor and outcome variables.**

| Baseline | All participants | | | Participants lost to follow-up | | | Participants who completed the follow-up | | | | | |
|---|---|---|---|---|---|---|---|---|---|---|---|---|
| Socio-demographics | N | M | SD | N | M | SD | N | M | SD | t | p | d |
| Age (years) | 201 | 38.55 | 10.13 | 104 | 37.45 | 10.55 | 97 | 39.73 | 9.59 | -1.60 | .11 | -0.23 |
| Relationship status (single = 1, relationship = 2) | 189 | 1.80 | 0.40 | 97 | 1.81 | 0.39 | 92 | 1.78 | 0.41 | 0.54 | .59 | 0.08 |
| Education (1–7) | 188 | 3.15 | 1.51 | 95 | 3.14 | 1.55 | 93 | 3.17 | 1.48 | -0.16 | .87 | -0.02 |
| Children (yes = 1; no = 2) | 190 | 1.57 | 0.50 | 97 | 1.59 | 0.49 | 93 | 1.55 | 0.50 | 0.54 | .59 | 0.08 |
| Weight (in kg) | 201 | 78.98 | 14.29 | 104 | 79.21 | 13.50 | 97 | 78.74 | 15.15 | 0.23 | .82 | 0.03 |
| Body Mass Index (kg/m$^2$) | 201 | 25.78 | 3.63 | 104 | 25.81 | 3.59 | 97 | 25.74 | 3.68 | 0.14 | .89 | 0.02 |
| Gender (male = 0, female = 1) | 201 | 0.36 | 0.48 | 104 | 0.39 | 0.49 | 97 | 0.32 | 0.47 | 1.10 | .27 | 0.15 |
| Shift work (yes = 0, no = 1) | 191 | 0.57 | 0.50 | 96 | 0.53 | 0.50 | 95 | 0.60 | 0.49 | -0.96 | .34 | -0.14 |
| Smoking (no = 0, yes = 1) | 189 | 0.20 | 0.40 | 96 | 0.19 | 0.39 | 93 | 0.20 | 0.41 | -0.29 | .77 | -0.03 |
| Drinking days per week | 189 | 1.33 | 1.38 | 96 | 1.34 | 1.35 | 93 | 1.31 | 1.42 | 0.16 | .87 | 0.02 |
| Physical activity and CF | N | M | SD | N | M | SD | N | M | SD | t | p | d |
| Accelerometry (MVPA min/week) | 171 | 410.63 | 174.3 | 86 | 403.27 | 154.21 | 85 | 418.18 | 193.13 | -0.56 | .58 | -0.09 |
| CF (estimated VO$_2$max in ml/kg/min) | 200 | 45.02 | 11.22 | 104 | 43.99 | 10.93 | 96 | 46.14 | 11.48 | -1.35 | .18 | -0.19 |
| Work stress | N | M | SD | N | M | SD | N | M | SD | t | p | d |
| JDC ratio | 190 | 0.96 | 0.19 | 97 | 0.94 | 0.19 | 93 | 0.98 | 0.20 | -1.21 | .23 | -0.21 |
| ERI ratio | 190 | 0.89 | 0.27 | 97 | 0.86 | 0.23 | 93 | 0.93 | 0.31 | -1.55 | .12 | -0.26 |
| Metabolic syndrome | N | M | SD | N | M | SD | N | M | SD | t | p | d |
| Waist circumference (cm) | 199 | 91.07 | 11.27 | 103 | 91.00 | 11.32 | 96 | 91.14 | 11.27 | -0.08 | .94 | -0.01 |
| TG (mmol·L−1) | 201 | 1.69 | 1.17 | 104 | 1.63 | 1.04 | 97 | 1.75 | 1.30 | -0.74 | .46 | -0.10 |
| HDL-C (mmol·L−1) | 200 | 1.82 | 0.40 | 104 | 1.80 | 0.41 | 96 | 1.84 | 0.40 | -0.65 | .52 | -1.00 |
| SBP (mm Hg) | 201 | 129.57 | 13.17 | 104 | 129.14 | 13.29 | 97 | 130.04 | 13.09 | -0.48 | .63 | -0.07 |
| DBP (mm Hg) | 201 | 85.10 | 10.25 | 104 | 84.96 | 10.07 | 97 | 85.26 | 10.49 | -0.21 | .84 | -0.03 |
| HbA1c (%) | 201 | 5.45 | 0.29 | 104 | 5.46 | 0.30 | 97 | 5.43 | 0.28 | 0.67 | .50 | 0.10 |
| Overall MetS score | 201 | 1.16 | 0.94 | 104 | 1.15 | 0.95 | 97 | 1.19 | 0.96 | -0.16 | .87 | -0.04 |

M = Mean; SD = Standard Deviation; d = Effektstärke Cohen's d; Skew = Skewness; Kurt = Kurtosis; MetS (%) = Percentage of participants that met the specific criterion of Metabolic Syndrome; TG = Triglyceride; HDL-C = High-Density Lipoprotein Cholesterol; SBP = Systolic Blood Pressure; DBP = Diastolic Blood Pressure; HbA1c = Glycated Hemoglobin; MetS = Metabolic Syndrome; JDC = Job Demand and Control; ERI = Effort-Reward Imbalance; MVPA = Accelerometer-based Moderate to Vigorous Physical Activity; CF = Cardiorespiratory Fitness. Differences in N are due to different numbers of missing values for different variables. Sample size is lower for MVPA than CF because some participants had to be excluded from data analyses due to insufficient accelerometer wear-time.

model. These sociodemographic variables explained 10 percent of variance in the overall MetS score in both models. The job stress questionnaires did not significantly explain additional variance in either of the models. Although both MVPA (β = -.17) and CF (β = -.38) were negatively associated with the overall MetS score, only the association for CF reached significance ($p < .01$), with CF explaining 12 percent of variance in the final model.

Fig 1 illustrates the significant interaction between the two work stress questionnaires in the cross-sectional analysis. With an increase in the ERI ratio, the group with a higher JDC ratio showed a greater increase in the overall MetS score.

The prospective hierarchical linear regression analyses are presented in Table 5. The level of explained variance in the dependent variable (overall MetS score) varied between 38 and 39 percent. Counter to the cross-sectional analyses, no significant interaction effects occurred in the prospective analyses. This indicates that differences in MetS at follow-up were not influenced by the interplay between baseline scores of job stress, physical activity, and CF. In all three models, relationship status significantly predicted the overall MetS score at follow-up.

**Table 2. Descriptive statistics for main study variables at baseline and follow-up for the participants who took part in the follow-up data assessment (*N* = 97).**

| Baseline | | | | | | | |
|---|---|---|---|---|---|---|---|
| Metabolic syndrome | *n* | *M* | *SD* | Range | Skew | Kurt | MetS (%) |
| Waist circumference (cm) | 96 | 91.14 | 11.27 | 61.50–126.00 | 0.48 | 1.26 | 20 (20.6) |
| TG (mmol·L$^{-1}$) | 97 | 1.75 | 1.30 | 0.51–7.35 | 2.52 | 7.10 | 34 (35.1) |
| HDL-C (mmol·L$^{-1}$) | 96 | 1.84 | 0.40 | 0.93–2.59 | 0.02 | -0.54 | 2 (2.1) |
| SBP (mm Hg) | 97 | 130.04 | 13.09 | 107.00–172.0 | 0.59 | 0.44 | 59 (60.8) |
| DBP (mm Hg) | 97 | 85.26 | 10.49 | 63.50–118.0 | 0.21 | 0.04 | |
| HbA1c (%) | 97 | 5.43 | 0.28 | 4.90–6.80 | 1.65 | 6.36 | 1 (1.0) |
| Overall MetS score | 97 | 1.19 | 0.96 | 0.00–4.00 | 0.55 | 0.07 | |
| | | | | | | | |
| Work stress | | | | | | | |
| JDC ratio | 93 | 0.98 | 0.20 | 0.54–1.50 | 0.60 | 0.25 | |
| ERI ratio | 93 | 0.93 | 0.31 | 0.33–2.02 | 0.79 | 1.01 | |
| | | | | | | | |
| Physical activity and CF | | | | | | | |
| Accelerometry (MVPA min/week) | 85 | 418.18 | 193.13 | 49–1389 | 1.77 | 6.91 | |
| CF (estimated VO$_2$max in ml/kg/min) | 96 | 46.14 | 11.48 | 24.20–89.40 | 0.69 | 1.09 | |
| | | | | | | | |
| Follow-up | | | | | | | |
| Metabolic syndrome | | *M* | *SD* | Range | Skew | Kurt | MetS (%) |
| Waist circumference (cm) | 96 | 90.87 | 10.90 | 72.0–127.0 | 0.94 | 1.43 | 18 (18.6) |
| TG (mmol·L$^{-1}$) | 97 | 1.84 | 0.98 | 0.7–7.4 | 2.69 | 11.05 | 47 (48.5) |
| HDL-C (mmol·L$^{-1}$) | 96 | 1.73 | 0.56 | 0.9–5.5 | 3.34 | 20.64 | 5 (5.2) |
| SBP (mm Hg) | 97 | 127.46 | 13.49 | 103.0–172.0 | 0.71 | 0.73 | 47 (48.5) |
| DBP (mm Hg) | 97 | 82.41 | 10.03 | 62.5–112.5 | 0.61 | 0.45 | |
| HbA1c (%) | 96 | 5.23 | 0.92 | 4.8–6.8 | 0.95 | 0.82 | 2 (2.1) |
| Overall MetS score | 97 | 1.22 | 0.92 | 0–4 | 0.92 | -0.11 | |

*M* = Mean; *SD* = Standard Deviation; Skew = Skewness; Kurt = Kurtosis; MetS (%) = Percentage of participants that met the specific criterion of Metabolic Syndrome; TG = Triglyceride; HDL-C = High-Density Lipoprotein Cholesterol; SBP = Systolic Blood Pressure; DBP = Diastolic Blood Pressure; HbA1c = Glycated Hemoglobin; MetS = Metabolic Syndrome; JDC = Job Demand and Control; ERI = Effort-Reward Imbalance; MVPA = Accelerometer-based Moderate to Vigorous Physical Activity; CF = Cardiorespiratory Fitness. Differences in *N* are due to different numbers of missing values for different variables. Sample size is lower for MVPA than CF because some participants had to be excluded from data analyses due to insufficient accelerometer wear-time.

Being in a relationship was negatively associated with the overall MetS score after one year. Furthermore, the stepwise inclusion of baseline values of the overall MetS score significantly explained between 12 and 14 percent of variance in the two models, showing that higher MetS scores at baseline were associated with higher MetS scores at follow-up. By contrast, the work stress measures did not significantly predict the overall MetS score after one year. MVPA was negatively associated with the overall MetS score after one year (β = -.18). However, the association did not reach statistical significance. By contrast, CF (β = -.25, p < .05) was significantly and negatively associated with the overall MetS score after one year, explaining 9 percent of variance in the model. This association is presented in Fig 2.

## Differences between fitness categories in MetS

Based on the significant main effects of CF in both the cross-sectional and prospective analyses, we provide further information describing this association. The following figures show the

**Table 3. Correlations between physical activity, cardiorespiratory fitness, work stress with cardiometabolic risk factors at baseline and follow-up, for the participants who took part in the follow-up data assessment.**

| Baseline | MVPA | CF | JDC ratio | ERI ratio | Demand | Control | Effort | Reward |
|---|---|---|---|---|---|---|---|---|
| Waist circumference (cm) | -.21 | -.39** | .06 | .13 | .18 | .06 | .19 | -.02 |
| n | 84 | 95 | 92 | 92 | 92 | 92 | 92 | 92 |
| TG (mmol·L⁻¹) | -.04 | -.26** | -.02 | .07 | .16 | .15 | .09 | -.05 |
| n | 85 | 96 | 93 | 93 | 93 | 93 | 93 | 93 |
| HDL-C (mmol·L⁻¹) | .01 | .28** | -.16 | -.11 | -.13 | .09 | -.13 | .04 |
| n | 84 | 95 | 92 | 92 | 92 | 92 | 92 | 92 |
| SBP (mm Hg) | -.21 | -.18 | -.05 | .20 | .11 | .12 | .25* | -.08 |
| n | 85 | 96 | 93 | 93 | 93 | 93 | 93 | 93 |
| DBP (mm Hg) | -.21 | -.25* | -.05 | .19 | .12 | .15 | .24* | -.05 |
| n | 85 | 96 | 93 | 93 | 93 | 93 | 93 | 93 |
| HbA1c (%) | -.27* | -.34** | -.12 | .05 | -.01 | .13 | .01 | -.05 |
| n | 85 | 95 | 93 | 93 | 93 | 93 | 93 | 93 |
| Overall MetS score | -.14 | -.38** | -.05 | .10 | .12 | .16 | .17 | .02 |
| n | 85 | 96 | 93 | 93 | 93 | 93 | 93 | 93 |
| | | | | | | | | |
| Follow-up | MVPA | CF | JDC ratio | ERI ratio | Demand | Control | Effort | Reward |
| Waist circumference (cm) | -.17 | -.41** | .02 | .20 | .23* | .17 | .26* | -.06 |
| n | 84 | 95 | 92 | 92 | 92 | 92 | 92 | 92 |
| TG (mmol·L⁻¹) | -.15 | -.19 | .00 | .10 | .16 | .10 | .07 | -.11 |
| n | 85 | 96 | 93 | 93 | 93 | 93 | 93 | 93 |
| HDL-C (mmol·L⁻¹) | .08 | .27** | .02 | .07 | .14 | .08 | -.01 | -.13 |
| n | 85 | 96 | 93 | 93 | 93 | 93 | 93 | 93 |
| SBP (mm Hg) | -.13 | -.19 | -.04 | .19 | .13 | .13 | .23* | -.08 |
| n | 85 | 96 | 93 | 93 | 93 | 93 | 93 | 93 |
| DBP (mm Hg) | -.12 | -.25** | -.12 | .17 | .08 | .21* | .21* | -.08 |
| n | 85 | 96 | 93 | 93 | 93 | 93 | 93 | 93 |
| HbA1c (%) | -.29** | -.26* | -.16 | .00 | -.08 | .08 | -.08 | -.09 |
| n | 84 | 95 | 92 | 92 | 92 | 92 | 92 | 92 |
| Overall MetS score | -.21* | -.38** | -.10 | .10 | .08 | .19 | .12 | -.03 |
| n | 85 | 96 | 93 | 93 | 93 | 93 | 93 | 93 |

MVPA = Accelerometer-based Moderate-to-Vigorous Physical Activity; CF = Cardiorespiratory Fitness; JDC = Job Demand and Control; ERI = Effort-Reward Imbalance; TG = Triglyceride; HDL-C = High-Density Lipoprotein Cholesterol; SBP = Systolic Blood Pressure; DBP = Diastolic Blood Pressure; HbA1c = Glycated Hemoglobin; MetS = Metabolic Syndrome. Differences in $N$ are due to different numbers of missing values for different variables. Sample size is lower for MVPA than CF because some participants had to be excluded from data analyses due to insufficient accelerometer wear-time.

\* $p < .05$;

\*\* $p < .01$.

differences between fitness levels as classified by the ACSM in regard to their overall MetS score at baseline (Fig 3A) and follow-up (Fig 3B).

## Discussion

The aim of the present paper was to assess whether physical activity and cardiorespiratory fitness moderate the interplay between work stress and MetS. We followed 97 police officers for one year. Our results revealed no significant stress-buffering effects of physical activity or cardiorespiratory fitness. However, higher levels of CF significantly predicted lower levels of

**Table 4. Cross-sectional hierarchical linear regression with overall MetS score at baseline as dependent variable, for the participants who took part in the follow-up data assessment.**

| | Stress-buffering variable | | | |
| --- | --- | --- | --- | --- |
| | MVPA (*n* = 82) | | CF (VO$_2$max) (*n* = 92) | |
| | $\Delta R^2$ | β | $\Delta R^2$ | β |
| Step 1[a] | .10* | | .10** | |
| Age | | .15 | | .15 |
| Education | | -.28* | | -.22* |
| Step 2 | .01 | | .01 | |
| JDC ratio | | -.11 | | -.02 |
| ERI ratio | | -.02 | | .05 |
| Step 3 | .02 | | .12** | |
| Stress-buffering variable | | -.17 | | -.38** |
| Step 4 | .05 | | .07* | |
| JDC ratio x ERI ratio | | .21 | | .25** |
| Stress-buffer x JDC ratio | | .07 | | .04 |
| Stress-buffer x ERI ratio | | -.19 | | -.17 |
| Total $R^2$ | .18* | | .30** | |

MVPA = Accelerometer-based Moderate-to-Vigorous Physical Activity; CF = Cardiorespiratory Fitness; JDC = Job Demand and Control; ERI = Effort-Reward Imbalance. [a]Only covariates were retained in the final model for which a significant bivariate association was found in the correlation analyses. Differences in *N* are due to different numbers of missing values for different variables. Sample size is lower for MVPA than CF because some participants had to be excluded from data analyses due to insufficient accelerometer wear-time.

* $p < .05$;

** $p < .01$.

MetS in the regression analyses. Neither job stress, nor physical activity were significant predictors of MetS after one year. We formulated three hypotheses, which will now be discussed in turn.

In our first hypothesis, we expected that work stress would be associated with MetS. Whereas the two job stress questionnaires cross-sectionally correlated with the outcome as expected, these correlations only reached significance for subscale values. The strongest correlations occurred for waist circumference, and blood pressure. This is in line with previous evidence showing that the association between job stress and MetS was mediated by obesity [54]. However, we did not find a significant direct effect between the JDC ratio or ERI ratio and MetS in the regression analyses. This result does not match with the strong associations reported in previous studies. Data from the Whitehall study showed that chronic work stress (job strain and effort-reward imbalance) over 14 years was related to an increased risk for MetS of 2.25 OR (95% confidence interval = 1.31 to 3.85) [14]. Another example is a review of prospective cohort studies with an approximately 8-year follow-up by Siegrist [55]. In this review, the risk for Type II diabetes was estimated to be 160 percent for high effort-reward imbalance [55]. Shorter follow-up durations, however, have shown weaker associations. In a two-year follow-up study by Loerbroks et al. [15], job stress (ERI) was associated with an adjusted risk for MetS of 1.20; with a 95 percent confidence interval of 1.01 to 1.42. Hence, the duration of the present study, but also the low incidence of MetS, might have contributed to lower detectable effects.

The above-mentioned correlations between job stress and the components of MetS in the present study mainly applied for the demand (Job Content Questionnaire) and effort (Effort-

**Table 5. Prospective hierarchical linear regression with overall MetS scores at follow-up as dependent variable, for the participants who took part in the follow-up data assessment.**

| | Stress-buffering variable | | | |
| --- | --- | --- | --- | --- |
| | MVPA ($n = 81$) | | CF (VO$_2$max) ($n = 91$) | |
| | $\Delta R^2$ | β | $\Delta R^2$ | β |
| Step 1[a] | .20** | | .15** | |
| Relationship status | | -.33** | | -.25** |
| Education | | -.11 | | -.11 |
| Step 2 | .12** | | .14** | |
| Overall MetS score at baseline | | .37** | | .33** |
| Step 3 | .01 | | .01 | |
| JDC ratio | | -.05 | | -.06 |
| ERI ratio | | .00 | | -.01 |
| Step 4 | .02 | | .09** | |
| Stress-buffering variable | | -.18 | | -.25* |
| Step 5 | .03 | | .01 | |
| JDC ratio x ERI ratio | | -.17 | | -.09 |
| Stress-buffer x JDC ratio | | -.05 | | .07 |
| Stress-buffer x ERI ratio | | -.03 | | -.05 |
| Total $R^2$ | .38** | | .39** | |

MetS = Metabolic Syndrome; MVPA = Accelerometer-based Moderate-to-Vigorous Physical Activity; CF = Cardiorespiratory Fitness; JDC = Job Demand and Control; ERI = Effort-Reward Imbalance. [a]Only covariates were retained in the final model for which a significant bivariate association was found in the correlation analyses. Differences in $N$ are due to different numbers of missing values for different variables. Sample size is lower for MVPA than CF because some participants had to be excluded from data analyses due to insufficient accelerometer wear-time.

\* $p < .05$;

\*\* $p < .01$.

Reward Imbalance Questionnaire) subscales. Garbarino and Magnavita [20] found similar results in a sample of highly stressed police officers. When looking at the separate subscales in a five-year prospective study, only demand and effort were significant predictors of MetS. Furthermore, we found an unexpected positive correlation between the control subscale and some components of MetS, which needs further consideration. In this respect, we want to introduce a model by Carayon and Zijlstra [56]. This model posits different influences on job strain by three dimensions of job control; namely task control, resource control, and organizational control. Control over tasks performed and resources used are considered to negatively influence job strain. Organizational control, however, might show different effects. Organizational control can include group and organizational responsibilities. The term 'active job', by Karasek and Theorell [57], inheres in this category. While people in such positions can delegate tasks, they remain responsible for, and accountable to the success of the outcome. Carayon and Zijlstra [56] argued that work pressure, and hence job strain, would increase with such organizational control. Our results are, therefore, important to highlight for future research. To the best of our knowledge, no study has examined health-related effects of different control dimensions in police officers. Further distinctions might enhance the quality of evidence for the Job Content Questionnaire, as this is one of the most frequently applied job stress questionnaires.

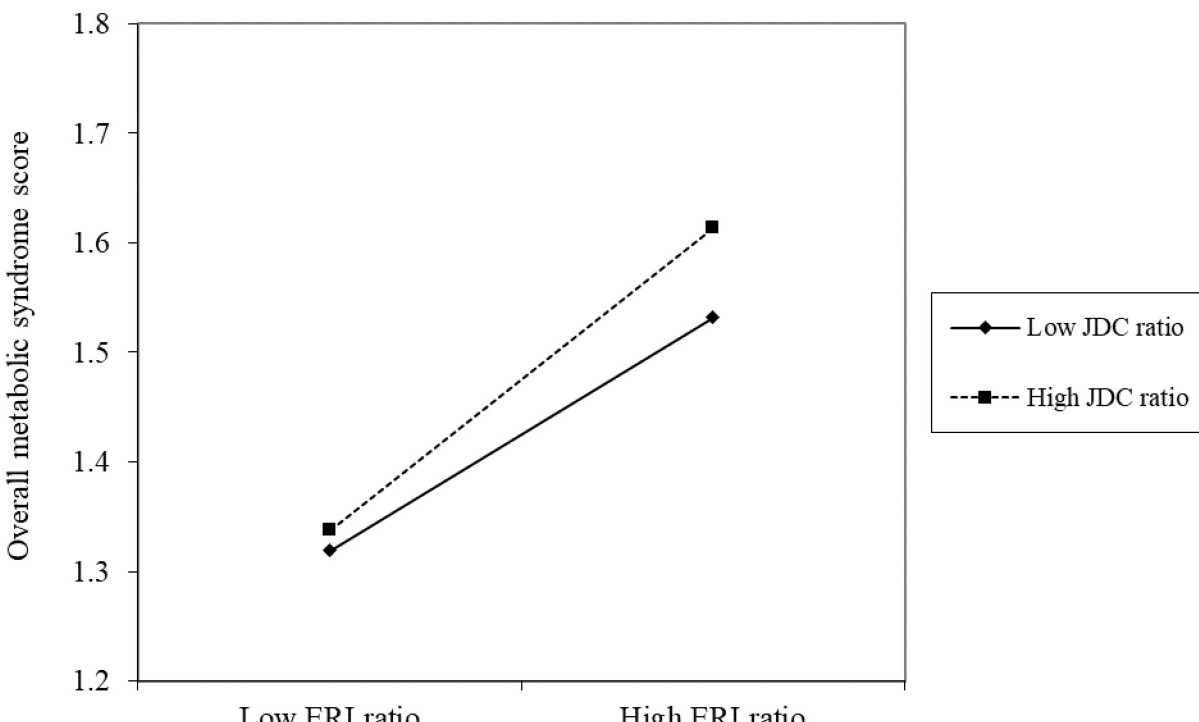

**Fig 1. Graphical representation of the interaction between JDC ratio and ERI ratio in predicting overall MetS scores at baseline ($n$ = 82).**

With our second hypothesis, we expected an inverse relationship between physical activity or CF and MetS. Our results fully support this hypothesis for CF, whereas only partial support was found for physical activity. Whereas in the correlational analyses, higher levels of physical activity were associated with more favorable scores in some (but not all) MetS components, no significant association was found between MVPA and the overall MetS score in the regression analyses. In contrast, several previous studies reported relatively strong correlations between physical activity and Type II diabetes, blood pressure, obesity, and lipid levels [58–60]. For the interpretation of our findings, it is important to consider international recommendations for MVPA, such as those of the World Health Organization (WHO) or the American Heart Association [53, 61]. These recommendations highlight that adults should accumulate at least 150 minutes of weekly MVPA. Although higher physical activity levels are considered to be even more health-enhancing, the greatest health benefits are thought to occur in the shift from inactivity to recommended levels [61]. In a Canadian sample which was described as physically active, 23.9 percent of the 2324 participants reached recommended levels of MVPA [62], and the study results yielded strong correlations between MVPA and MetS [62]. Compared to this sample, where only a minority of participants accomplished recommended MVPA standards, our sample was much more active, with 97 percent meeting the recommended minimum of 150 minutes of MVPA per week. Hence, the possible health benefits for increased physical activity levels might be limited (ceiling effect). In line with this notion, some scholars have argued that it is likely to find the strongest relationships between physical activity and MetS in physically inactive and unfit individuals [63]. In summary, we suggest that the high overall physical activity level of our study participants has lowered the potential to detect a significant relationship between MVPA and MetS.

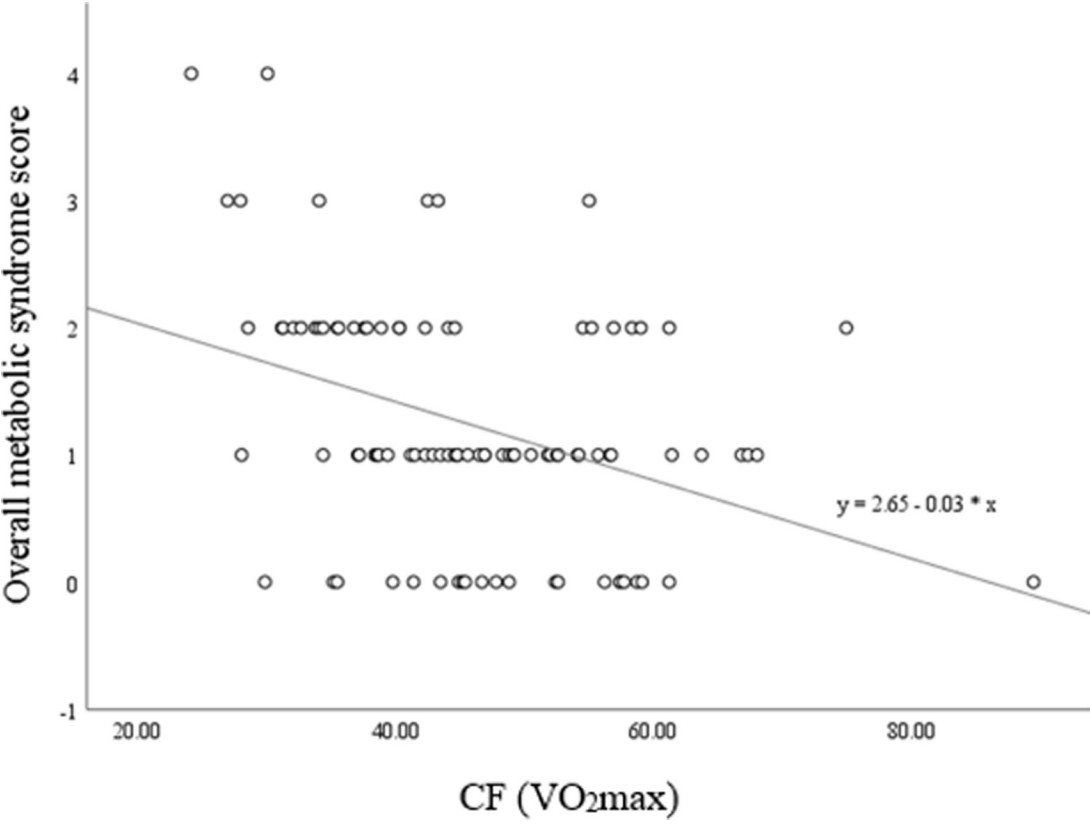

**Fig 2. Scatterplot with line of best fit capturing the association between CF (VO₂max) levels and overall MetS scores at follow-up (*n* = 96).**

In spite of this, we found full support for our hypothesis with regard to CF. Throughout all analyses (correlations; regression analyses) higher levels of CF were associated with more favorable scores in almost all of the MetS components. In cross-sectional analyses, only the association between CF and blood sugar and systolic blood pressure did not reach the level of significance. Furthermore, CF significantly predicted the overall MetS score after one year, explaining 9 percent of variance in the final statistical model. This association vividly showed in the group differences based on the ACSM classifications of cardiorespiratory fitness (Fig 3A and 3B). Participants with very poor to poor fitness showed higher overall MetS scores compared to the rest of the sample. This difference appeared more distinct regarding the follow-up values of overall MetS scores. These results are in line with previous evidence from the general population, and the occupation of law enforcement specifically [64, 65]. We also want to emphasize the importance of these results in view of the aforementioned relevance of cardio-vascular diseases and mortality, with MetS being suggested as a potential link between work stress and cardiovascular diseases [13].

With our third hypothesis, we expected a moderation effect of physical activity or CF on the relationship between work stress and MetS. This hypothesis seemed plausible as previous investigations mostly supported stress-buffering effects of physical activity and CF [25, 66]. These effects have been ascribed to different physiological pathways including the hypotha-lamic-pituitary-adrenal (HPA) axis and the sympatho-adrenal medullary (SAM) axis [67]. While MetS is not regarded a proxy of these axes, research has shown a greater interest in

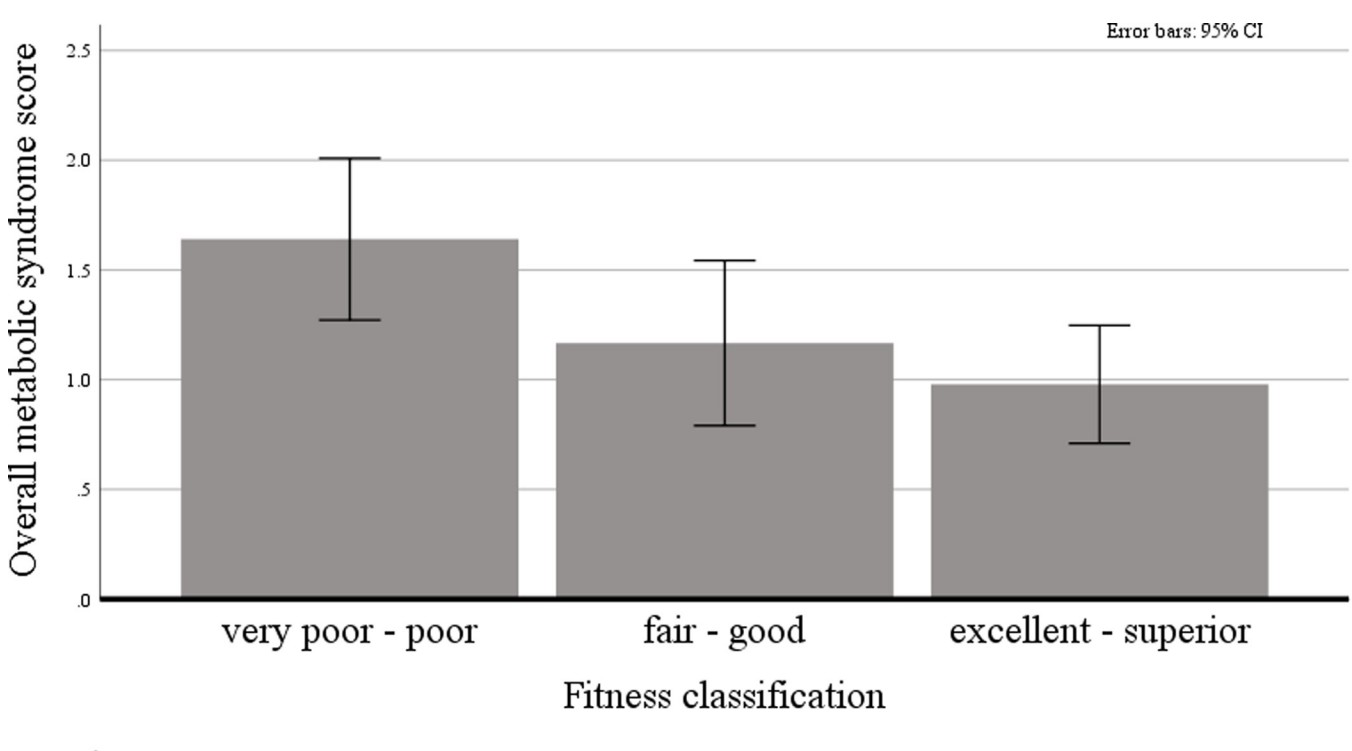

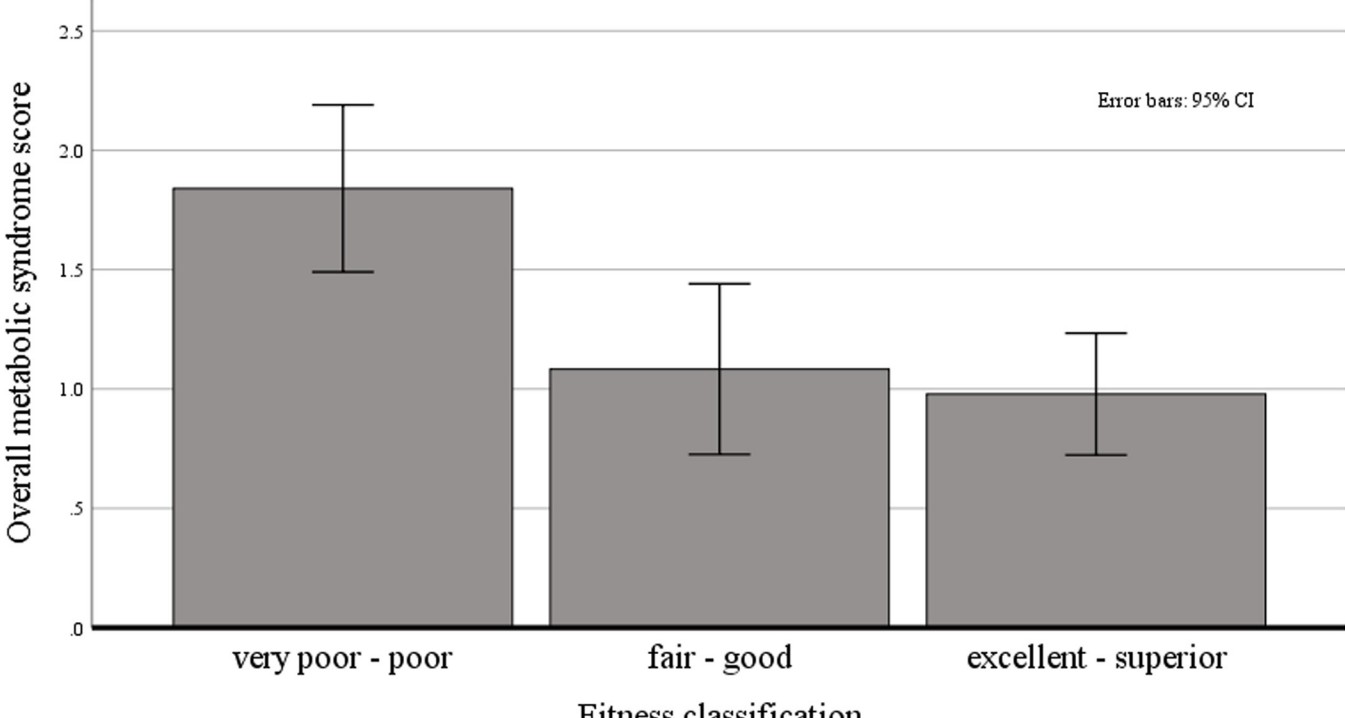

**Fig 3. A.** Bar plot with confidence intervals (95%) regarding the distribution of overall MetS scores at baseline in the different fitness classifications following ASCM guidelines (*n* = 96). **B.** Bar plot with confidence intervals (95%) regarding the distribution of overall MetS scores at follow-up in the different fitness classifications following ASCM guidelines (*n* = 96).

closely related parameters [68]. In this regard, cortisol or further inflammatory risk markers such as C-reactive protein are important factors associated to cardiovascular disease [7, 69]. In

line with this notion, Violanti et al. [70] observed a significant shift in the cortisol response in highly stressed police officers. Nevertheless, in a study by Franke et al. [71], work stress (job strain and effort-reward imbalance) did not predict increased inflammatory risk markers in police officers. Moreover, it is also conceivable that psychosocial stress has an impact on health behaviors (e.g. nutrition) which might contribute to the occurrence of MetS [25], including epigenetic or biochemical variations [72, 73]. For example, Charles et al. [74] showed that physical activity significantly interacted with oxidative stress and obesity in a sample of police officers.

Despite these solid theoretical foundations, in the present sample of police officers, the interactions of physical activity and CF with job stress did not significantly predict MetS after one year. Only very few studies have tested stress-buffering effects of physical activity and CF in police officers. Gerber et al. [19] showed that exercise levels and fitness buffered the effects of work stress on health in a sample of Swiss police officers. However, these results relied on self-reports of fitness, exercise, and perceived health. In an earlier study, Young [75] included 412 law enforcement officers. Using a cross-sectional design, Young [75] did not find significant moderation effects of cardiorespiratory fitness in the interplay between stress and risk factors for coronary artery disease. Similarly to our study, Young [75] did not find significant direct associations between job stress and cardiovascular risk factors. Moreover, only few previous studies looked at cardiometabolic risk factors. For instance, in a study with Swedish health-care workers, Gerber et al. [27] observed that higher stress scores were associated with an increased overall cardiometabolic risk. Under these conditions, Gerber et al. [27] showed that participants with high stress who also had high CF levels had lower scores for systolic and diastolic blood pressure, LDL cholesterol, triglycerides and total cardiometabolic risk than participants with high stress but low or moderate CF levels. Similar stress-buffering effects on specific MetS components were found in child and adolescent samples [28]. A recent systematic review described the evidence regarding the association between work stress and cardiovascular risk factors in police officers [76]. Although the association appears to be of a positive nature, Magnavita et al. [76] highlighted the importance of longitudinal studies with large sample sizes in order to find significant effects. Given that work stress did not significantly correlate with components of MetS in the present study, the potential for physical activity and CF to moderate this relationship was limited in the present analysis. Thus, we hold that the lack of significant main effects of job stress might be one of the primary reasons we did not find a stress-buffering effect in the present population.

The strengths of our study are the prospective research design and the fact that objective data was assessed for physical activity (7-day accelerometry), CF (submaximal fitness test) and cardiometabolic risk factors. A lack of objective assessments has recently been identified as one of the key limitations in this area of research [27, 77]. The only self-report variable used in the present study was work stress. However, since stress is personal experience based on cognitive-transactional appraisal processes, it is difficult to find an objective indicator. While potential biomarkers such as hair cortisol are discussed in the literature [78], the validity of such indicators remains questionable in predominantly healthy populations [79, 80]. We have therefore decided to use well-accepted and validated tools to assess job stress, which have been previously employed with police officers [76]. In the assessment of cardiometabolic risk, we followed one of the most widely accepted definitions for MetS. To the best of our knowledge, no study has examined the direct association between objectively assessed physical activity and MetS in police officers.

Despite these strengths, the generalizability of our results might be limited due to several reasons. First, our sample was highly active, and most officers had relatively high CF levels. Furthermore, prevalence of MetS in the present sample was low. While in previous studies

with police officers in the United States, prevalence rates for MetS ranged between 16 and 26 percent [17, 24, 32, 33], the prevalence was considerably lower in our sample (7% at baseline, 8% at follow-up). In a review of the literature, Yoo et al. [81] concluded an overall lowered risk for MetS in US police officers compared to the general publication. Although no populations-based data exist for Switzerland, a comparison with 12 cohorts from 10 European countries (N = 34'821 subjects), in which the overall prevalence of MetS was 24.3 percent (23.9% in men, 24.6% in women), shows that the level of MetS was far below average in our study. The prevalence rate found in our study must be interpreted with caution, as participation in the investigation was voluntary. Thus, in the sense of a healthy worker effect, it is likely that more healthy officers were more willing to participate in the study. Furthermore, based on the increased standard error related to the statistical analyses, the present sample size may have been a limiting factor in detecting effects. Finally, the follow-up period (one year) was relatively short in the present investigations, given that most studies finding significant associations between work stress and MetS followed participants across much longer periods of time [14, 55, 82].

## Conclusion

MetS is increasing at an alarming rate in industrialized countries. The prevalence rates and close link to cardiovascular disease and mortality have generated a great interest in research of preventive and rehabilitative factors. Although the prevalence of MetS was relatively low in the present sample, our results highlight the importance of CF in the prevention of MetS in highly active and fit individuals. Accordingly, provision of regular training opportunities and repeated CF testing should be considered as a strategy in overall corporate health promotion.

## Acknowledgments

We would like to thank all participants of the police corps of the Canton of Basel-City for taking part in our study. We further want to express our gratitude to the management of the police corps for their support and the possibility to use police internal infrastructures throughout the entire study. In this respect, all employees involved in the recruitment, administration, and facilitation processes deserve special thanks for their essential efforts in implementing the study. Finally yet importantly, we thank the master students Benjamin Grossmann and Nico Güdel for their contribution to the data assessment. The research endeavor complies with the current laws of the country in which they were performed.

## Author Contributions

**Conceptualization:** René Schilling, Uwe Pühse, Markus Gerber.

**Data curation:** René Schilling, Markus Gerber.

**Formal analysis:** René Schilling, Markus Gerber.

**Investigation:** René Schilling.

**Methodology:** René Schilling, Flora Colledge, Uwe Pühse, Markus Gerber.

**Project administration:** René Schilling, Uwe Pühse.

**Resources:** Uwe Pühse.

**Software:** René Schilling, Markus Gerber.

**Supervision:** Markus Gerber.

**Validation:** René Schilling.

**Visualization:** René Schilling, Flora Colledge, Markus Gerber.

**Writing – original draft:** René Schilling, Flora Colledge, Markus Gerber.

**Writing – review & editing:** René Schilling, Flora Colledge, Uwe Pühse, Markus Gerber.

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
