## [Decision Letter · Decision Letter 0]

2 Jun 2020

PONE-D-20-08673

Stress-Buffering Effects of Physical Activity and Cardiorespiratory Fitness on Metabolic Syndrome: a Prospective Study in Police Officers

PLOS ONE

Dear Dr. Schilling

Thank you for submitting your manuscript to PLOS ONE. After careful consideration, we feel that it has merit but does not fully meet PLOS ONE’s publication criteria as it currently stands. Therefore, we invite you to submit a revised version of the manuscript that addresses the points raised during the review process.

We look forward to receiving your revised manuscript.

Kind regards,

Matteo Vandoni

Academic Editor

PLOS ONE

Journal Requirements:

Additional Editor Comments (if provided):

Manuscript has major issues to correct. Please provide appropriate responses to the reviewers

Reviewers' comments:

Reviewer's Responses to Questions

**Comments to the Author**

1. Is the manuscript technically sound, and do the data support the conclusions?

Reviewer #1: Yes

Reviewer #2: Partly

2. Has the statistical analysis been performed appropriately and rigorously? 

Reviewer #1: Yes

Reviewer #2: Yes

3. Have the authors made all data underlying the findings in their manuscript fully available?

Reviewer #1: Yes

Reviewer #2: Yes

4. Is the manuscript presented in an intelligible fashion and written in standard English?

Reviewer #1: Yes

Reviewer #2: Yes

5. Review Comments to the Author

Reviewer #1: This manuscript analyses direct associations of physical activity (PA), cardiorespiratory fitness (CRF) and two models of work stress (JDC, ERI) with metabolic syndrome (MetS) both cross-sectionally and over 12 months. In addition, moderation effects of PA, CRF, JDC and ERI are tested in a hierarchical linear regression model. The prospective study contains 97 police officers. Results indicate a significant prospective association of CRF with MetS. Additionally, several main and interaction effects with subcomponents are observed, without providing clear support of the research hypotheses.

The manuscript has several strengths, including the prospective design, the well-described, mostly objective measures, the careful data analysis, and an extensive discussion of results, with substantial reference to the state of art. Despite these strengths, a number of queries become evident from this submission:

1. Design and data analysis: In the Methods section, authors mention that the laboratory test was performed twice during a year, in addition to a follow-up assessment of MetS after 12 months. However, in the Results section, all respective data refer to baseline assessment. It is thus not clear how data of a second wave were treated. It is also not clear whether work stress was assessed twice. For instance, in Table 1, when follow-up data were presented, the values for work stress were exactly identical with those of cross-sectional analysis. Authors need to clarify their procedure.

2. Sample: The sample is defined by 97 prospective participants. However, authors eventually mention that 201 police officers participated in the cross-sectional analysis. It is not clear if cross-sectional results were systematically related to the larger sample and prospective results to the smaller sample. Unfortunately, authors did not describe the sample size in the tables. It is mandatory to mention the sample size in all Tables and Figures!

3. Follow-up: Authors did not describe the initial sample size of invited police officers. What was the response rate? Moreover, they did not analyse attrition from 201 to 97 participants at follow-up. It is important to know more about selection (attrition) factors in order to interpret the comparative findings from the cross-sectional and prospective associations.

4. Outdated references: Although it does not concern the main research hypothesis, the references given for prospective associations of work stress with CHD (Ref. 39, 42, 43) are outdated. There is substantial new evidence asvailable. For JCQ see a s revew:

Kivimäki M, Steptoe A (2018) Effects of stress on the development and progression of cardiovascular disease. Nat Rev Cardiol 15(4):215. For ERI see recent publication: Dragano N, Siegrist J, Nyberg ST et al.: Effort-Reward Imbalance at work and incident coronary heart disease: a multicohort study of 90,164 individuals. Epidemiology 28 (2017), S. 619-626.

5. Moderation: Authors should be cautious when testing interaction effects with a sample lower than 100 participants. It is highly unlikely to observe robust results. At least, this should be discussed as a limitation.

6. Minor: It may be problematic to introduce CRF as abbreviation for cardiorespiratory fitness as CRF is an established abbreviation for Corticotropin-releasing factor in biomedical research. Another detail: Garbarino et al. is mentioned in Introduction, but the reference is only given in Discussion (Ref. 54).

Reviewer #2: The purpose of the manuscript “Stress-Buffering Effects of Physical Activity and Cardiorespiratory Fitness on Metabolic Syndrome: a Prospective Study in Police Officers” was to investigate potentially stress-buffering effects of CRF on MetS”. Unfortunately, this manuscript presents many limitations and weakness. The authors didn’t purpose any programme of physical activity and the text could be developed in different way. In particular:

Title: it could be changed in “Effects of work stress on Metabolic Syndrome in Police Officers: a Prospective Study”

Abstract: description of methods is uncomplete.

Line 22: put on age and BMI of 97 police officers.

Line 28-31 put on data of basal level and of follow up and of correlation/association

Introduction: Please revise completly. After I read the introduction, I asked to me: Is the aim of study innovative? It same no innovative…..

Line 79-89: I think that you can delete….You have three hypothesis that are showed by references 33, 34, and 25.

Discussion and conclusion: Please revise completly. They are same at the introduction about the innovation of this study…..difficult to see the "take-home" message in the text and the practical applications of study…..

6. PLOS authors have the option to publish the peer review history of their article (what does this mean?). If published, this will include your full peer review and any attached files.

Reviewer #1: Yes: Johannes Siegrist

Reviewer #2: No

---

## [Author Response · Author response to Decision Letter 0]

25 Jun 2020

Revision statement

Reviewer #1

This manuscript analyses direct associations of physical activity (PA), cardiorespiratory fitness (CRF) and two models of work stress (JDC, ERI) with metabolic syndrome (MetS) both cross-sectionally and over 12 months. In addition, moderation effects of PA, CRF, JDC and ERI are tested in a hierarchical linear regression model. The prospective study contains 97 police officers. Results indicate a significant prospective association of CRF with MetS. Additionally, several main and interaction effects with subcomponents are observed, without providing clear support of the research hypotheses.

The manuscript has several strengths, including the prospective design, the well-described, mostly objective measures, the careful data analysis, and an extensive discussion of results, with substantial reference to the state of art. Despite these strengths, a number of queries become evident from this submission:

Response: Thank you for the positive appraisal of our paper and for the constructive and specific feedback. We have addressed all issues and provided further clarification.

Comment 1: Design and data analysis: In the Methods section, authors mention that the laboratory test was performed twice during a year, in addition to a follow-up assessment of MetS after 12 months. However, in the Results section, all respective data refer to baseline assessment. It is thus not clear how data of a second wave were treated. It is also not clear whether work stress was assessed twice. For instance, in Table 1, when follow-up data were presented, the values for work stress were exactly identical with those of cross-sectional analysis. Authors need to clarify their procedure.

Response: Thank you very much for your statement. While data was assessed twice within one year, calculations encompassed follow-up data on MetS only. We discarded information on fitness and physical activity levels due to the research question. In this respect, we apologize for a mistake in Table 1. We accidently added work stress values, as well as physical activity and fitness components for follow-up assessments, which was very unfortunate and obviously misleading. We thank you very much for noticing this issue. We have deleted the respective data (Table 1 [revised Table 2]). Furthermore, we have added the assessed outcome variables to the description of the design (Line 100), as well as a statement on procedure to the statistical analyses section (Line 207).

Comment 2: Sample: The sample is defined by 97 prospective participants. However, authors eventually mention that 201 police officers participated in the cross-sectional analysis. It is not clear if cross-sectional results were systematically related to the larger sample and prospective results to the smaller sample. Unfortunately, authors did not describe the sample size in the tables. It is mandatory to mention the sample size in all Tables and Figures!

Response: Thank you very much for your statement. While trying to stay as concise as possible, we are keen to transfer all information necessary to clarify our procedure. All calculations were carried out including only participants who completed both measurements (n = 97). Accordingly, we have added a statement to the sample description section (Line 241). We further added the according sample sizes to all tables and all figure descriptions.

Comment 3: Follow-up: Authors did not describe the initial sample size of invited police officers. What was the response rate? Moreover, they did not analyse attrition from 201 to 97 participants at follow-up. It is important to know more about selection (attrition) factors in order to interpret the comparative findings from the cross-sectional and prospective associations.

Response: Thank you for your suggestion. We agree that this information is important. Approximately 1000 police force employees received the study advertisement, which they could view voluntarily. Unfortunately, we have no information about the number of police officers who watched the advertisements. Out of these 1000 possible recipients, 227 officers (approximately 23%) agreed to provide background information via the e-learning program, and 201 officers eventually decided to participate in the study (88%). In the revision, we have provided an additional table (Table 1) in order to depict possible selection factors. An error according to attrition seems unlikely, because based on the calculated t-tests, we did not find significant differences between participants lost to follow-up and completers in any of the sociodemographic background, predictor or outcome variables.

Comment 4: Outdated references: Although it does not concern the main research hypothesis, the references given for prospective associations of work stress with CHD (Ref. 39, 42, 43) are outdated. There is substantial new evidence available. For JCQ see as revew:

Kivimäki M, Steptoe A (2018) Effects of stress on the development and progression of cardiovascular disease. Nat Rev Cardiol 15(4):215. For ERI see recent publication: Dragano N, Siegrist J, Nyberg ST et al.: Effort-Reward Imbalance at work and incident coronary heart disease: a multicohort study of 90,164 individuals. Epidemiology 28 (2017), S. 619-626.

Response: Thank you very much for the suggested references. We agree that these are well suited, and have included them accordingly.

Comment 5: Moderation: Authors should be cautious when testing interaction effects with a sample lower than 100 participants. It is highly unlikely to observe robust results. At least, this should be discussed as a limitation.

Response: We appreciate your comment on the influence of sample size regarding the ability to detect possible interaction effects in the present analyses. Literature also suggests that the power of the applied tests in the analyses highly depends on the reliability and variability of the outcome (Beaujean, 2008), due to the increased standard error based on the interaction term. Therefore, a bigger sample size is indeed beneficial, especially since range restriction (relatively healthy sample) might have influenced the present results. We acknowledged this circumstance by adding a statement to the limitation section (Line 567).

“Furthermore, based on the increased standard error related to the statistical analyses, the present sample size may have been a limiting factor in detecting effects.”

Beaujean, A. A. (2008). Mediation, moderation, and the study of individual differences. In J. Osborne (Ed.), Best Practices in Quantitative Methods. Thousand Oaks, California: SAGE Publications, Inc

Comment 6: Minor: It may be problematic to introduce CRF as abbreviation for cardiorespiratory fitness as CRF is an established abbreviation for Corticotropin-releasing factor in biomedical research. 

Response: Thank you very much for your suggestion. We have changed the abbreviation to CF.

Comment 7: Another detail: Garbarino et al. is mentioned in Introduction, but the reference is only given in Discussion (Ref. 54).

Response: Again, thank you very much for noticing that mistake. We have added the relevant reference (Ref. 20; Line 70).

Reviewer #2

The purpose of the manuscript “Stress-Buffering Effects of Physical Activity and Cardiorespiratory Fitness on Metabolic Syndrome: a Prospective Study in Police Officers” was to investigate potentially stress-buffering effects of CRF on MetS”. Unfortunately, this manuscript presents many limitations and weakness.

Response: We thank the Reviewer for his/her time devoted to our manuscript and for suggesting specific changes how the quality of our paper can be improved. We have addressed all issues in our revision and hope that the revision was done to the satisfaction of the Reviewer. 

Comment 1:

The authors didn’t purpose any programme of physical activity and the text could be developed in different way.

Response: Thank you very much for your statement. We are not sure whether we have correctly understood your request. The approach of our study was observational, we did not intend to include a physical activity intervention. Furthermore, we did not recommend a physical activity programme in the conclusion section since our results did not show significant associations with MetS after controlling for sociodemographic characteristics (regression analysis). We would appreciate further specification on that matter if we did not understand correctly or provided misleading information.

Comment 2: In particular: Title: it could be changed in “Effects of work stress on Metabolic Syndrome in Police Officers: a Prospective Study”

Response: Thank you very much for your suggestion. We are not sure if we understand correctly. The study was set up to test stress-buffering or moderation effects of physical activity (PA) and cardiorespiratory fitness (CF). We built this argument by assembling the supporting evidence for the negative health effects of MetS, the association of psychosocial stress and MetS, as well as the stress-buffering potential of PA and CF. We highlighted the public health relevance of these components and that police officers are particularly at risk for stress and MetS. We have referenced the literature which supports these statements. Given the argumentation, design, and methodology, we regard PA and CF, as well as the assumed interaction effect, as key components of our research endeavour with a relevance for public health. PA and CF were treated as main variables in the introduction, in all analyses, as well as in the discussion. Therefore, we are confident that their appearance in the title is appropriate and helpful for readers to understand the content of the manuscript. Furthermore, several studies in this research area which did not apply behavioural interventions used the moderating variables prominently in the title (see references below). If we did not understand the suggestion correctly, or did not address important information in this response, we would kindly ask you for more background information.

References with the moderator variable in the title: 

Carmack, C. L., Boudreaux, E., Amaral-Melendez, M., Brantley, P. J., & de Moor, C. (1999). Aerobic fitness and leisure physical activity as moderators of the stress-illness relation. Annals of Behavioral Medicine, 21(3), 251-257.

Puterman, E., Lin, J., Blackburn, E., O'Donovan, A., Adler, N., & Epel, E. (2010). The power of exercise: Buffering the effect of chronic stress on telomere length. Plos One, 5(5), e10837.

Teisala, T., Mutikainen, S., Tolvanen, A., Rottensteiner, M., Leskinen, T., Kaprio, J., . . . Kujala, U. M. (2014). Associations of physical activity, fitness, and body composition with heart rate variability-based indicators of stress and recovery on workdays: a cross-sectional study. J Occup Med Toxicol, 9, 16.

Schmidt, K.-H., Beck, R., Rivkin, W., & Diestel, S. (2016). Self-control demands at work and psychological strain: The moderating role of physical fitness. International Journal of Stress Management, 23, 255-275.

Strahler, J., Doerr, J. M., Ditzen, B., Linnemann, A., Skoluda, N., & Nater, U. M. (2016). Physical activity buffers fatigue only under low chronic stress. Stress, 19(5), 535-541.

Comment 3: Abstract: description of methods is uncomplete.

Line 22: put on age and BMI of 97 police officers.

Response: Thank you very much for your suggestion. We have added information regarding the age and BMI of the sample to the abstract (Line 24), and to the sample description (Line 247).

Comment 4: Line 28-31 put on data of basal level and of follow-up and of correlation/association

Response: Again, thank you for your suggestion. We have added mean levels for the overall MetS scores at baseline and follow-up (Lines 30 and 31). Furthermore, we have added the coefficients for the regression and correlation (Line 32-36) analyses mentioned in the abstract.

Comment 5: Introduction: Please revise completely. After I read the introduction, I asked to me: Is the aim of study innovative? It same no innovative…..

Response: Thank you for your suggestion. We appreciate the suggestion to highlight the innovative aspects of our study. However, we are not sure that we have understood the suggestion correctly. Therefore, we briefly want to paraphrase the reasons we have given in the text regarding the novelty of our study. First, we focused on the assessment of objective parameters. This methodology is novel, since previous research strongly relies on subjective data. Second, we highlighted that stress and MetS are global societal/public health issues while evidence on PA and CF as potential stress-buffers is lacking. Third, we conducted a prospective study, which has been widely called for in the current literature. Therefore, we believe that the innovative aspects of our study have been discussed in the manuscript. 

Comment 6: Line 79-89: I think that you can delete….You have three hypothesis that are showed by references 33, 34, and 25.

Response: Thank you very much for your suggestion. We are not sure if we understand correctly. Deleting Line 79-89 would mean that we discard our study hypotheses. We are confident that the presentation of the hypotheses is important for the reader and would politely ask for further information about why we should delete these.

Comment 7: Discussion and conclusion: Please revise completly. They are same at the introduction about the innovation of this study…..

Response: Thank you for your statement. We agree that we partly repeated the innovations of our study in the discussion section. However, we did that with a focus on quality criteria of our data assessment in order to discuss the strengths of the study.

Comment 8: difficult to see the "take-home" message in the text and the practical applications of study…..

Response: Thank you very much for your observation. We agree that the practical implication/take-home message could be mentioned more clearly. We therefore condensed the conclusion section and added a statement on the importance of incorporating cardiorespiratory fitness testings in corporate health promotion (Line 577).

---

## [Decision Letter · Decision Letter 1]

9 Jul 2020

Stress-Buffering Effects of Physical Activity and Cardiorespiratory Fitness on Metabolic Syndrome: A Prospective Study in Police Officers

PONE-D-20-08673R1

Dear Dr. Schilling,

We’re pleased to inform you that your manuscript has been judged scientifically suitable for publication and will be formally accepted for publication once it meets all outstanding technical requirements.

Kind regards,

Matteo Vandoni

Academic Editor

PLOS ONE

Reviewers' comments:

Reviewer's Responses to Questions

**Comments to the Author**

1. If the authors have adequately addressed your comments raised in a previous round of review and you feel that this manuscript is now acceptable for publication, you may indicate that here to bypass the “Comments to the Author” section, enter your conflict of interest statement in the “Confidential to Editor” section, and submit your "Accept" recommendation.

Reviewer #1: All comments have been addressed

2. Is the manuscript technically sound, and do the data support the conclusions?

Reviewer #1: Yes

3. Has the statistical analysis been performed appropriately and rigorously? 

Reviewer #1: Yes

4. Have the authors made all data underlying the findings in their manuscript fully available?

Reviewer #1: Yes

5. Is the manuscript presented in an intelligible fashion and written in standard English?

Reviewer #1: Yes

6. Review Comments to the Author

Reviewer #1: I have read the revised Version of the manuscript and all Reviewer comments and respective answers by authors. All queries that I raised in my review have been resolved by authors. As far as I could understand the comments of Reviewer 2, authors tried best to answer them and to offer convincing reasons for their improved manuscript.

Overall, this is a carefully conducted study with relevant new Information on a Topic of considerable public Health relevance. The Quality of the study is high, and the study meets all ethical criteria.

7. PLOS authors have the option to publish the peer review history of their article (what does this mean?). If published, this will include your full peer review and any attached files.

Reviewer #1: **Yes: **Johannes Siegrist

---

## [Editor Report · Acceptance letter]

13 Jul 2020

PONE-D-20-08673R1 

Stress-Buffering Effects of Physical Activity and Cardiorespiratory Fitness on Metabolic Syndrome: A Prospective Study in Police Officers 

Dear Dr. Schilling:

I'm pleased to inform you that your manuscript has been deemed suitable for publication in PLOS ONE. Congratulations! Your manuscript is now with our production department. 

Kind regards, 

on behalf of

Dr. Matteo Vandoni 

Academic Editor

PLOS ONE